# Using high-resolution contact networks to evaluate SARS-CoV-2 transmission and control in large-scale multi-day events

Rachael Pung [1,2,3✉], Josh A. Firth [4,5], Lewis G. Spurgin [6], Singapore CruiseSafe working group*, CMMID COVID-19 working group*, Vernon J. Lee[1,7] & Adam J. Kucharski[2,3]

The emergence of highly transmissible SARS-CoV-2 variants has created a need to reassess the risk posed by increasing social contacts as countries resume pre-pandemic activities, particularly in the context of resuming large-scale events over multiple days. To examine how social contacts formed in different activity settings influences interventions required to control Delta variant outbreaks, we collected high-resolution data on contacts among passengers and crew on cruise ships and combined the data with network transmission models. We found passengers had a median of 20 (IQR 10–36) unique close contacts per day, and over 60% of their contact episodes were made in dining or sports areas where mask wearing is typically limited. In simulated outbreaks, we found that vaccination coverage and rapid antigen tests had a larger effect than mask mandates alone, indicating the importance of combined interventions against Delta to reduce event risk in the vaccine era.

[1] Ministry of Health, Singapore, Singapore. [2] Centre for the Mathematical Modelling of infectious Diseases, London School of Hygiene and Tropical Medicine, London, UK. [3] Department of Infectious Disease Epidemiology, London School of Hygiene and Tropical Medicine, London, UK. [4] Department of Zoology, University of Oxford, Oxford, UK. [5] Merton College, University of Oxford, Oxford, UK. [6] School of Biological Sciences, University of East Anglia, Norwich, UK. [7] Saw Swee Hock School of Public Health, National University of Singapore, Singapore, Singapore. *Lists of authors and their affiliations appear at the end of the paper. ✉email: rachael.pung@lshtm.ac.uk

Many countries are resuming domestic activities as vaccination coverage and population immunity against SARS-CoV-2 increases[1–3]. Settings with particularly high contact rates, such as meetings, conferences, exhibitions, and cruises, are also revenue-generating sectors with high pre-pandemic visitor throughput across the world[4,5]. However, the transmission dynamics on real world networks of large-scale events are yet to be fully explored in the COVID-19 era[6]. Furthermore, while pre-COVID-19 studies on human contact networks for understanding the transmission of infections spread by close contacts have analysed various network properties and attempted to reconstruct the social network from contact diaries or digital sensors, they are largely focused in school, healthcare settings or the greater community, with few studies on conferences and business meetings[7–10]. Understanding the risk of outbreaks in these settings and possible outbreak control interventions would enable event planners to gauge the sustainability of their operations and for policy makers to weigh the public health cost against the economic gains. Given breakthrough infections in vaccinated individuals and the spread of the highly transmissible SARS-CoV-2 variants[11–13], countries have employed a range of tools alongside routine vaccination to suppress disease transmission, including vaccine certifications, rapid antigen tests, mask mandates, and digital contact tracing devices[14,15]. Although there have been efforts to estimate infection risk during large events from routine testing data and contact tracing interviews[16], data from contact tracing devices can enable finer-scale assessment of interactions such as the distance and duration of contact depending on the strength and continuity of the Bluetooth signals captured in these devices. Furthermore, these devices overcome the challenges of recall bias and achieve more reliable estimates of the contacts in a network[17].

In Singapore, 'cruises to nowhere' (i.e. cruises that depart and return to the port of origin without other ports of call) began as a safe travelling option during the COVID-19 pandemic with a range of activities and hence setting-specific interactions onboard. We collected contact data from around 1000 crew and 1300 passengers per sailing between November 2020 and February 2021 and analysed the resulting social interaction networks. We then use these contact networks to simulate SARS-CoV-2 Delta variant outbreaks and assess how different combinations of interventions and network formulations influence transmission in a range of settings during a large-scale event.

## Results

**Characterising social interactions on cruise ships**. 3,963,256 contact episodes with 1,846,312 unique contact pairs were recorded during 37-h data collection periods across four separate three-day sailings (see Methods). During the period studied, cruise lines were operating at 50% capacity with a passenger to crew ratio of approximately 1:1 and passengers from different travelling groups were strongly advised to maintain a physical distance of at least one metre from other groups.

The four sailings had a mean of 1304 passengers (range 1142–1682) with a median age of 54 (IQR 35–63) and a mean of 1050 crew (range 1003–1083) spread across eight work departments (Table 1). There was a high density of contacts among passengers, with some clustering of contacts among the crew, although crew members may be required to work with other individuals from the same or different departments, and roles such as housekeeping and galley crew had contacts dispersed across the network (Fig. 1a and Supplementary Fig. 1). The crew was encouraged to form 'work bubbles' as part of COVID-19 workplace interventions (i.e. team of workers that work independently from another team). As a result, on average they

| Table 1 Demographics of passengers and department allocation of crew onboard four cruise sailings. | |
|---|---|
| **No. of passengers = 1304 (1142–1682)** | |
| **Demographics** | |
| Median age across all sailings in years (IQR) | 54 (35–63) |
| Passengers by age group | |
| <12 years | 47 (36–61) |
| 12–29 years | 166 (123–285) |
| 30–39 years | 184 (99–327) |
| 40–49 years | 164 (144–199) |
| 50–59 years | 285 (268–317) |
| 60–69 years | 314 (274–336) |
| ≥70 years | 146 (95–176) |
| Gender | |
| Female | 676 (602–832) |
| Male | 625 (540–850) |
| | |
| **No. of crews = 1050 (1003–1083)** | |
| **Department[a]** | |
| Entertainment | 77 (73–81) |
| Food & Beverage (F&B) | 179 (171–185) |
| Galley | 214 (208–219) |
| Gaming | 175 (163–187) |
| Hotel services | 84 (77–92) |
| Housekeeping | 123 (114–137) |
| Marine | 154 (148–160) |
| Security | 44 (40–48) |

Number of passenger and crew presented as mean with range in brackets, unless specified otherwise.
[a]Details of each department are provided in Supplementary Table 2.

had 10 unique close contacts per day (IQR 6–18), about 50% lower than that of passengers (median 20, IQR 10–36) (Fig. 1b). If the threshold for close contact (defined as a cumulative duration of the interaction of 15 min in our baseline analysis) was relaxed to a shorter duration, the overall median unique close contacts scaled exponentially (Fig. 1c). The strength of each contact (i.e. edge weights) can be further quantified as a function of the duration of the contact (see Methods). Adjusted for the duration of each contact, the median weighted degree in crew was 8.3 (IQR 4.4–13.5), while the median in passengers was 13.9 (IQR 5.6–23.7) (Table 2). Furthermore, passengers had significantly higher connectivity with other highly connected individuals, with a median eigenvector centrality of 0.3 (IQR 0.1–0.5) compared to a median of 0.09 (IQR 0.03–0.2) for the crew (Table 2 and Supplementary Fig. 2).

**Analysing the contacts formed during activities**. The total number of contacts made by passengers with passengers from other travelling groups varied according to the type of location and the time spent at that location. The total close contacts plateaued at approximately 3 (IQR 2–5) after spending at least 1 h in a food and beverage (F&B) location (Fig. 2a) while the total close contacts were 2 (IQR 1–3) after spending 30 min to 1 h in a sports location and increased to 4 (IQR 2–7) after spending at least 2 h (Fig. 2c).

Over the three-day sailings, a median of 71% (IQR 64–74%) of all the close contact episodes occurring between passengers from different travelling groups occurred in F&B locations of which 23% (IQR 19–26%) and 38% (IQR 31–40%) occurred in the buffet and inclusive restaurants respectively (Fig. 3a, b). 16% (IQR 11–24%) of the close contacts occurred in entertainment areas and 8% (IQR 6–10%) in sports areas (Fig. 3a). Passengers are largely mask-off when dining or engaged in sports and this

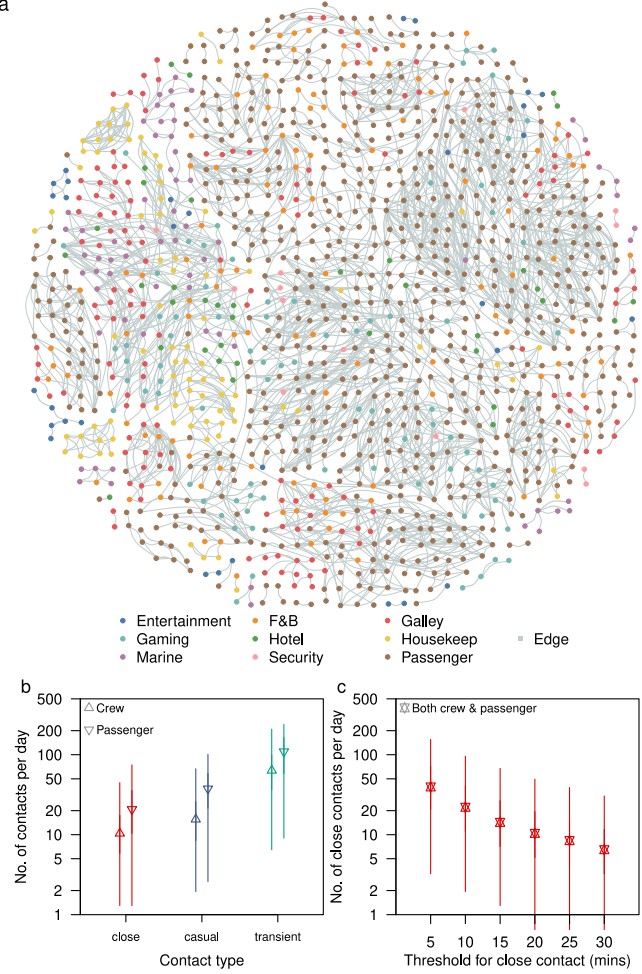

**Fig. 1 Distribution of cruise ship contacts. a** Illustrative short-term network dynamics, showing the cumulative network of all contacts that began between 12.00 to 12.05 pm on the second day of a sail and lasted till the end of their contact episode. Edge width and colour intensity are a function of the type of contact (i.e. close, casual and transient). Intra- and inter-cohort contacts are represented by the connection of nodes with the same and different colour respectively. **b** Number of unique close, casual, transient contacts made per day by passenger and crew. **c** Number of close contacts per day for both crew and passengers using different thresholds for the cumulative duration of interaction. Median (shapes), 50% (dark lines), and 95% intervals (light lines) of contacts from 5216 passengers and 4197 crew across four sailings are shown in (**b**) and (**c**).

accounted for 79% (IQR 69–84%) of all close contact episodes, 69% (IQR 57–76%) causal, and 60% (IQR 51–66%) transient contact episodes (Fig. 3a).

**Modelling outbreak dynamics and interventions**. To examine the spread of SARS-CoV-2 on cruise ships and implications for other large-scale multi-day events, we used the contact data to generate an undirected network with nodes and edges representing individuals and the contact between them respectively. We defined the strength of an edge as a function of the proportion of days with recorded contact over a three-day sail period and the mean daily cumulative contact duration between two individuals to approximate a scenario where the edge weight reached 95% saturation after 3 h of contact (see Methods). This meant that the propensity for transmission increased and stabilised after 3 h of contact, to mimic contacts formed in family gatherings over extended periods of time[18,19].

**Table 2 Network properties of passengers and crew onboard four cruise sailings.**

| Network properties | Passenger | Crew | *P*-value |
|---|---|---|---|
| Weighted degree | 13.9 (5.6–23.7) | 8.3 (4.4–13.5) | $<2.2 \times 10^{-16}$ |
| Eigenvector centrality | 0.3 (0.1–0.5) | 0.08 (0.03–0.2) | $<2.2 \times 10^{-16}$ |
| Clustering coefficient | 0.4 (0.3–0.4) | 0.3 (0.2–0.4) | $<2.2 \times 10^{-16}$ |

Two sided Welch's *t*-test was performed and results were presented as median with IQR in brackets.

We extended a community network transmission model[20,21] to simulate SAR-CoV-2 Delta variant transmission over seven days (Table 3), to enable comparison between different interventions during early generations of transmission. We considered interventions including: (i) one-off PCR testing one day before the sailing (to allow for test turnaround time), (ii) rapid antigen testing at the start and halfway through the event, (iii) mask wearing in feasible settings and (iv) vaccination coverage among attendees. In both (i) and (ii) testing interventions, we assumed infected individuals were isolated immediately after a positive test in the main analysis. The sensitivity of the PCR and rapid antigen tests were assumed to vary with viral load modelled according to the Delta variant[12,22–24]. For the mask wearing intervention, we assumed that passengers of different travelling groups would be exposed to each other without a mask during dining, sports activities (e.g. pool and waterslides, rock climbing, basketball, football) or smoking breaks; and would be wearing a mask correctly otherwise. Furthermore, contacts between passengers and crew were assumed to occur with mask-on at all times and crew-crew contacts were assumed to occur without a mask during meals times, workouts or smoking breaks. The proportion of contacts that occurred without a mask were modelled based on the proportion of contacts occurring in F&B and sport settings (Fig. 3a).

Under the baseline scenario with no modelled interventions, with one initial infected individual and assuming that the event lasted for 7 days, we estimated a median of 10 individuals (IQR 3–23) would be infected by the end of the event (Fig. 4a and Supplementary Fig. 3a). Of these, 90% (IQR 84–100) would only develop symptoms after the event. Because presymptomatic transmission was assumed to account for 25% the transmission, more than two generations of infections could sometimes occur during the event (Supplementary Fig. 3b). We estimated that 64% and 17% of the simulated outbreaks involved spillover from passengers to crew and inter-department crew transmission respectively, and we estimated that spillover events first occurred in the 2nd (IQR 2–3) and 3rd (IQR 3–4) generation respectively. Outbreaks with a final size of more than 10 cases occurred in 48% of our simulations (Fig. 4b).

With the introduction of a one-off PCR test one day prior to the start of the cruise, the index case was isolated in 49% of the time, while 5% of the remaining simulations resulted in no transmission due to the stochastic nature of early disease transmission and the structure of the social network (Fig. 4b). As a result, more than half of the simulations had zero secondary cases. The risk of an outbreak of more than 10 cases was reduced to 22% with the PCR intervention. However, with rapid antigen testing at the start and at halfway through the event instead, only 3% of simulations resulted in a large outbreak.

We also modelled passenger-passenger interactions occurring under a mask-off setting ~60% of the time (based on the total transient, casual and close contacts in Fig. 2a) and assumed that

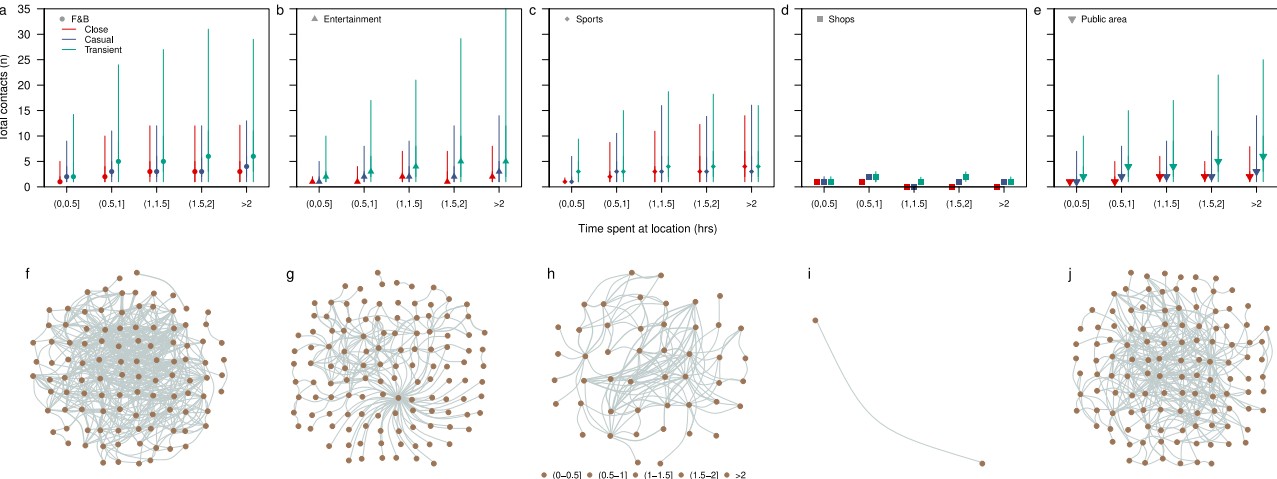

**Fig. 2 Number of contacts made over time in respective locations.** Contacts made between passengers from different travelling groups per visit to a type of location (**a–e**) and a snapshot of contact network at respective locations for 2 h intervals on the second day of the sailing (**f–j**). Type of locations are: F&B (**a**, **f**), entertainment (**b**, **g**), sports (**c**, **h**), shops (**d**, **i**) and public areas (**e**, **j**). Median (shapes), 50% (dark lines) and 95% intervals (light lines) of contacts from 5216 passenger and 4197 crew across four sailings are shown in (**a–e**). Nodes of different colour intensity represent the time spent in the location by respective passengers in (**f–j**).

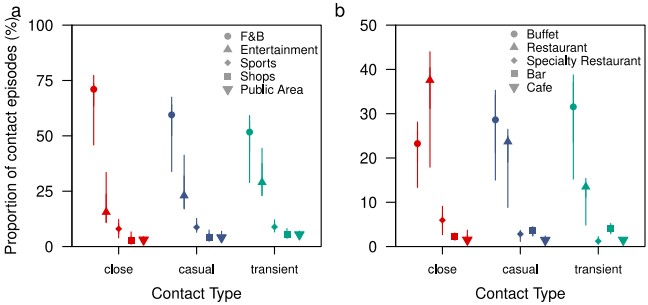

**Fig. 3 Type of contact by location of interaction and throughout the sailing. a** Proportion of all close, casual and transient contact episodes between passengers of different travelling groups by the location of interaction, **b** proportion of all close, casual, and transient contact episodes between passengers of different travelling groups by respective F&B locations. Median (shapes), 50% (dark lines) and 95% intervals (light lines) are shown.

all passenger-crew interactions occurred while wearing masks and that 30% of crew-crew interactions occurred when eating or working out without mask. Under these conditions and in the absence of other interventions, 22% of the simulations end with a large outbreak size (Fig. 4b). Assuming all individuals onboard the cruise ship were vaccinated (individuals under 12 years of age account for only 2% of the cruise population), 95% of simulations resulted in five or fewer cases (Fig. 4b).

We examined the expected outbreak size under a combination of interventions under the assumption that vaccination confers 50% protection against infection[13,25,26] and 50% lowered infectiousness in a vaccinated but infected individual[27] (Fig. 4c, d). Regardless of the testing strategies applied (i.e. no test, once-off PCR test, rapid antigen testing at the start and halfway through the event), and at any level of vaccine coverage, the addition of a mask-on intervention would further reduce the expected outbreak size by about 54% (IQR 50–59%). Given outbreak size is the cumulative result of individual transmission events, this implies that the overall intervention effectiveness of a mask mandate is substantially less than the assumed mask-on efficacy at the individual level (Table 4)[28–30]. The expected outbreak size in simulations involving rapid antigen testing was

<1 when vaccine coverage was minimally 25% (i.e. the expected number of transmission events was less than the initial number of infected individuals) (Fig. 4c). The expected outbreak size in mask-on, no testing interventions differed from mask-off, once-off PCR testing intervention by <1 case across varying vaccination coverage. The same was observed between a mask-on, once off PCR testing intervention and a mask-off, rapid antigen testing intervention. Compared to the expected outbreak size, the 95th percentile of the outbreak size is approximately three times higher, with the no testing, mask-off and one-off PCR testing, mask-off interventions generating the highest number of cases among all other combinations of interventions (Fig. 4d).

Sensitivity analysis under different assumptions of the edge weights—and hence per-contact risk—showed an increase in the expected outbreak size as the duration required to be defined as a 'maximal contact' (i.e. weight of one) decreases (Supplementary Fig. 4). Across all scenarios of varying testing strategy, vaccination coverage, network assumptions for edge weight, the average reduction in the expected outbreak size between a mask-on and mask-off scenario was 60% (IQR 54–71%). Assuming edge weights vary based on the proportion of days over the entire sailing when interactions were recorded (i.e. a transient contact in a day is as risky as a close contact in a day), the difference in the expected outbreak size between a mask-on, no testing scenario and a mask-off, once-off PCR testing widens to 32 cases (IQR 11–64) (Supplementary Fig. 4c). The difference in the expected outbreak size between a mask-on, once-off PCR testing scenario and a mask-off, rapid antigen testing at the start and halfway through the event differed by 6 cases (IQR 5–19) (Supplementary Fig. 4c). We obtained similar conclusions on the relative effect of different combinations of interventions when we varied assumptions about the extent of vaccine effectiveness and presymptomatic transmission (see Supplementary Information).

In reality, transmission parameters and effectiveness of outbreak interventions exhibit various uncertainties that can act simultaneously (Supplementary Table 1), and contact networks are temporally dynamic as the presence/absence of edges in the network change over time. Accounting for the uncertainty in the transmission process, our results for the expected and 95th percentile of the outbreak size falls between those in simulations assuming 25–50% presymptomatic transmission (Supplementary Figs. 4 and 6). As compared to the main analysis, the risk

**Table 3 Parameter values and assumptions.**

| Parameter | Assumed values | Details and references |
|---|---|---|
| Incubation period (days), $\theta$ | Lognormal distribution with Mean = 4.4, sd = 1.9 | [51] |
| Adherence to isolation when tested positive (%) | 100 | For scenarios involving testing only and we assume that there are available cabins for individuals to isolate given that cruises are operating at 50% capacity. |
| Delay from positive test to isolation (hrs) | No delay | For scenarios involving testing, individuals were isolated once tested positive. |
| Initial cases among passengers | 1 | |
| Scaling parameter, $r_{scale}$ | 0.24–0.26 | Each network formulation uses one scaling parameter value to calibrate the probability of Delta infection among cabin contacts to be similar to that of household contacts of 20%[34,49,52]. The range of values used across the different network formulations are as shown. |

reduction through a mask-on intervention has a wider uncertainty of 40–80% while the adherence to isolation after testing positive could be as low as 60%. As such, both interventions will perform lower but we observed narrower differences in outbreak size for a mask-on, no testing scenario and a mask-off, one-off PCR testing scenario (Supplementary Fig. 8). The lowered adherence to isolation coupled with the possibility of vaccinated infected individuals being as infectious as unvaccinated individuals resulted in a larger outbreak size observed in all testing interventions at low vaccine coverage. This was in spite of the potential for vaccinated susceptible contacts achieving a higher risk reduction against infection of 50–70% which counteracts the reduced effectiveness of the aforementioned interventions. When simulating outbreaks on a dynamic network, we accounted for the heterogeneity in the contact duration over the days and the sequence of contact episodes. As passengers engaged in more activities on the second day on the cruise sailing as compared to the first, the number of contacts and duration is correspondingly higher. A static network that averages out these heterogeneity in contact could thus allow for a higher potential of transmission in earlier stages of the cruise sailing, resulting in a higher 95th percentile as compared to a temporal network (Supplementary Figs. 9 and 10). Nevertheless, it is encouraging to note that the median outbreak size is similar for outbreaks simulated in both a static and temporal network (Supplementary Figs. 9 and 10) as simulations on longer time scales of 7 days were performed using the static network which served as a means of extending the network beyond 3 days.

For context, in real-life cruise operations during 2021, over 80% of the population received two doses of COVID-19 vaccination and a one-off pre-event rapid antign testing was required. No outbreaks occurred on these cruises even when the reported community incidence was 0.7 per 1000 at the height of the outbreak in end of October 2021—approximately 30% lower than that simulated in the model (i.e. one initial infected passenger corresponding to a community incidence of about 1 per 1000).

## Discussion

We found that the structure and intensity of contacts over a multi-day cruise have major consequences for outbreak control in different settings, particularly if there are mask-free activities and leaky testing protocols mean infectious individuals are likely to go undetected. Cruises represent an aggregation of different activities including F&B, entertainment, sports, meeting, conference, entertainment and workplace settings. The presence of multi-group passengers and crew from different departments can therefore offers insights into the potential dynamics of different actors in other large-scale multi-day events (e.g. a conference where there are participants, organising teams, external vendors, front-end and back-end F&B service staff, audiovisual support teams) and resulting implications for control of SARS-CoV-2.

Our social network analysis showed that passengers had a high number of contacts and their contacts typically exhibit high levels of contact with other individuals. As such, any disease transmission would likely be driven by passenger-level interactions rather than crew. In early 2020, this was evident in the sharp rise in the number of COVID-19 passengers with symptom onset before or during the early stages of quarantine onboard the Diamond Princess[31]. While the number of contacts made with other passengers are potentially lowered due to physical distancing and awareness of the pandemic in the studied Singapore setting, the number and type of activities onboard the cruise still means that each passenger forms around 20 unique close contacts per day. Compared to an average of 59 unique close contacts with more than 15 min of interaction in a UK community setting over a 14-day period[32], this was five times higher, further illustrating the intensity of contacts during such events. More than 70% of the close and casual contacts on the cruises occurred in F&B locations where passengers were largely mask-off and thus posing a higher risk of infection and transmission. We observed that the number of close contacts plateaued in F&B settings as the time spent in the location increases. As such, reducing this risk potentially requires more creative use of space to increase the distance between groups of passengers, improve indoor ventilation and encourage more outdoor dining.

With numerous work functions interfacing with passengers, and given the overlapping shifts and closely related job scope between crew (e.g. F&B and galley, hotel services and housekeeping), we found it only took around two generations for the infection to spread from a passenger to a crew and an additional generation of transmission to reach another crew in a different department. For SARS-CoV-2 transmission on the Diamond Princess cruise ship, the earliest onset in crew occurred about 18 days after the onset of the index case[33]. Assuming a generation time of about 5–7 days, this corresponds to a spillover from passengers to the crew after three to four generations of transmission. With about 2.6 times more passengers than crew on the Diamond Princess cruise ship, this could delay the spillover of disease transmission. Crew and event personnel play an important role in ensuring smooth operations and their wellbeing should be accounted for in the plans when reopening events. Hence, besides encouraging crew cohorting, interventions that minimise transmission in passengers would have an indirect effect of protecting the crew.

When applied individually, none of the interventions analysed were capable of reducing the expected outbreak size to be lower

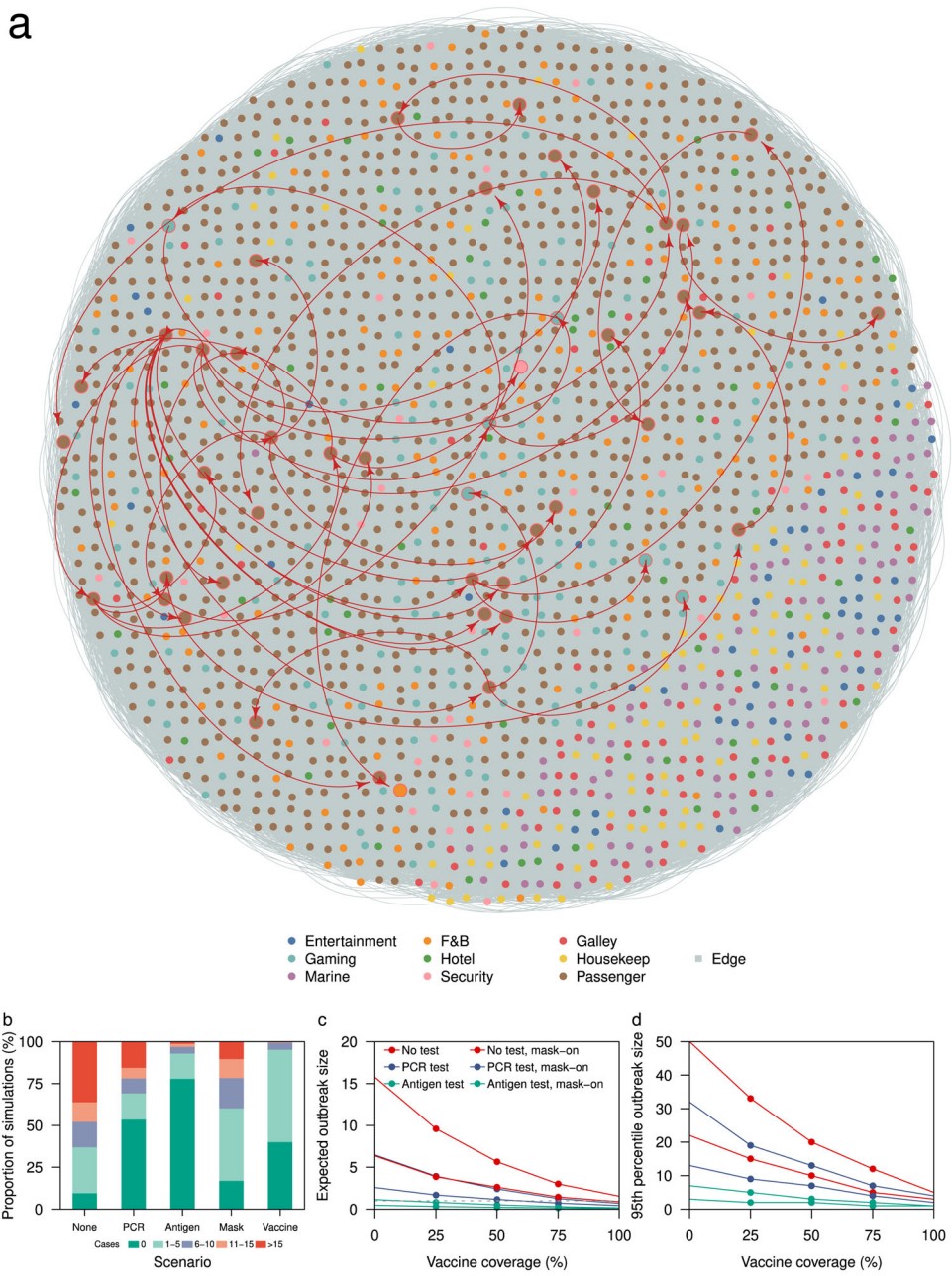

**Fig. 4 Outbreak size under respective interventions. a** Cases and contacts in one outbreak simulation with cases represented by an enlarged node and red curved arrows depicting disease transmission, **b** proportion of simulations by respective outbreak size under different interventions, **c** average outbreak size and **d** 95th percentile of outbreak size for different interventions and varying vaccination coverage.

than one; the number of initial infected cases in passengers, equivalent to a community incidence of 1 per 1000 individuals. However, a combination of rapid antigen testing at the start and halfway through the sailing with at least 25% coverage of a vaccine that confers 50% protection against infection and 50% lowered infectiousness would result in the cruise event having fewer onward transmission than the number of initial infectives. This is conditioned on the cases exhibiting Delta variant-like high viral loads with prolonged shedding[12,23,24,34] which improves the sensitivity of rapid antigen tests. While PCR tests have a higher sensitivity than rapid antigen tests at low viral load levels, the tests need to be conducted on land prior to the event due to the turnaround time required and for validity of lab results. This implies that cases who develop symptoms several days after the sailing may not be identified prior to the event, due to

viral loads near the limits of detection, and large outbreaks could occur.

The expected outbreak size under different combinations of interventions was sensitive to the assumptions of the network edge weights. When edges are weighted by the proportion of days with recorded interaction over a three-day sail period, two individuals with transient contact in a day are assumed to have the same risk as two individuals with close contact in a day. This assumption is applicable when the dominant mode of transmission is largely independent of the duration of contact (e.g. environmental or airborne transmission). There were five times more transient interactions than close contacts and these contacts are now equally at risk of infection. Thus, mask wearing would largely help to lower the risk of transmission and acquiring infection, and outperforms PCR testing or even twice antigen

**Table 4 Parameter values for the relative risk of infection, $\beta$.**

| Notation | Vaccination and mask wearing status | Relative risk | Remarks |
|---|---|---|---|
| $\beta^v$ | Mask-off (i.e. not wearing a mask) and both the infected and susceptible individuals are not vaccinated. | 1 | No change in probability of infection. |
| | Mask-off and infected individual $i$ is vaccinated. | 0.5 | Mean probability of transmitting infection reduces by 50%[27]. |
| | Mask-off and susceptible individual $j$ is vaccinated. | 0.5 | Mean probability of acquiring infection reduces by 50%[13,25,26]. |
| | Mask-off and both infected individual $i$ and susceptible individual $j$ are vaccinated. | 0.25 | Mean probability of infection reduces by 75%. Assumes the effect of vaccination on transmission and acquiring infection is independent. |
| $\beta^{mv}$ | Mask-on (i.e. wearing a mask) only. | 0.2 | Mean probability of infection reduces by about 80% when both the infected individual and susceptible contact are wearing a mask[28]. |
| | Mask-on and infected individual $i$ is vaccinated. | 0.1 | Mean probability of infection reduces by about 90%. Assumes the effect of vaccination and mask wearing on reducing the probability of transmitting infection is independent. |
| | Mask-on and susceptible individual $j$ is vaccinated. | 0.1 | Mean probability of infection reduces by 90%. Assumes the effect of vaccination and mask wearing on reducing the probability of acquiring infection is independent. |
| | Mask-on and both infected individual $i$ and susceptible individual $j$ are vaccinated. | 0.05 | Mean probability of infection reduces by 95%. Assumes the effect of vaccination and mask wearing on reducing the probability of transmission and acquiring infection is independent. |

testing interventions especially when the proportion of pre-symptomatic transmission is high during the early stage of viral shedding. Overall, these models estimate that mask wearing with passengers practising physical distancing could reduce transmission by about 54% under these settings, ~20–30% lower than the effectiveness of wearing a surgical mask in healthcare settings or in public areas after accounting for interactions in mask-off settings when dining or engaged in sports. This findings corroborates with behavioural surveys reporting 1.3–2 times higher risk of being infected when mask wearing in enclosed spaces is not practised[29]. The risk reduction from these model estimates are about five times higher than that reported in a cluster-randomised trial. However, in this trial, proper mask wearing occurred in less than half in the intervention arm, thereby limiting the multiplicative effectiveness of mask wearing in reducing both infection and transmission[30].

Both mask-off, rapid antigen testing and mask-on, once-off PCR testing would help to reduce the risk of disease introduction and further transmission if the index cases successfully escape initial detection. While their differences in the expected outbreak size were less than 10 cases across different assumptions to the edge weights, they bring different outcomes to the passenger experience and operations planning—an extra swab test at the middle of the event versus wearing a mask at all times other than during dining and engaged in sports, logistics to check the test outcomes versus monitoring mask wearing practises, managing false positives versus passengers flouting rules. Pre-event rapid antigen testing has been widely adopted in many large-scale events lasting less than a day and accounted for about 53% reduction in transmission in settings with high levels of social contacts and about 72% reduction after accounting for physical distancing[2]. In a fully susceptible cohort, these models estimate a mask-off, rapid antigen testing intervention at the start and midway of the event would reduce the mean outbreak size by over 90% with the additional reduction largely attributed to the administration of an additional test midway through the event.

One limitation to our study was that we did not model contact tracing around detected cases and the behaviour of contacts who are aware of their potential exposures. Thus, our estimates serve as an upper bound to the potential outbreak size. While cruise lines are trained to trace and quarantine close contacts as part of the pilot reopening, as the ease of rapid testing increases with fast turnaround time, this could serve as a replacement for slower and resource intensive contact tracing in such settings. With pre-symptomatic transmission of SARS-CoV-2 and high levels of transmissibility of the Delta variant, the effectiveness of contact tracing is approaching a point of saturation in many countries[35]. Furthermore, even if the threshold for close contact to be traced is lowered, the corresponding exponential increase in contacts fulfilling this criteria would make it logistically challenging to trace all individuals in a reasonable amount of time. Fully asymptomatic infections—as opposed to presymptomatic infections—were also not considered in the analysis. Should these infections exhibit lowered viral load, the testing interventions would be less likely to detect asymptomatic individuals but any potential for increase in outbreak size would be counteracted by their lowered infectiousness. Currently, there is no strong evidence to suggest that asymptomatic SARS-CoV-2 Delta infections are less infectious than symptomatic individuals[36,37]. Lastly, the accuracy of the data collected is largely dependent on the usage behaviour and the functionality of the device. Passengers are required to carry the contact tracing devices at all times except when engaged in water sports, and this was enforced by crew and external officers. Hence, interactions at the water sports areas may not be well represented but this effect to our analysis is expected to be minor as the cruise line of study required passengers to book these facilities in advance to facilitate crowd control. In a cabin, each passenger's device may not necessarily be placed in a 2 m proximity and the frequent close contact interactions in these settings would not be recorded accurately. However, given that individuals in the cabin would largely continue to interact with each other outside the cabin while carrying the device, this would help to record a large proportion of their close interaction. Furthermore, this limitation is reduced when the probability of infection saturates after a certain level of exposure as is in the case of SARS-CoV-2[38]. Functional issues of the contact tracing device such as drainage of batteries and incomplete data uploading can affect the extent of missing data, but these issues can be minimised with proper training on device usage. In our main analysis, the chosen sailing had more than 97% coverage in both crew and passengers to minimise the impact of missing data on the inference of the outbreak dynamics. Outbreaks were also simulated in three other sailings as part of sensitivity analysis and similar trends in the outbreak trajectory were observed (Supplementary Fig. 7). Despite such limitations, this is one of the few studies with large- and fine-scale data collection from multiple events in one setting and comparison with future

studies of similar data collection methods in similar and other settings may help to strengthen our findings and provide better understanding of transmission dynamics under different network structure and disease characteristics.

Given the spread of highly transmissible SARS-CoV-2 variants alongside increasing vaccination coverage, many countries have oscillated between reopening and restrictions of varying extents, in turn affecting the sustainability of economic and social activities. As the pressure to resume large-scale events increases, but the effectiveness of vaccines against infection and transmission remains variable, combining social interaction data with models such as the one presented here can enable an improved data-driven assessment of the risk of transmission arising from planned activities and the potential reduction offered by the continuation and implementation of non-pharmaceutical interventions.

## Methods

**Ethics statement**. Information was provided and consent was obtained from all participants in the study before the digital contact tracing device recorded any data. The study was approved by the London School of Hygiene & Tropical Medicine Observational Research Ethics Committee (ref. 25727).

**Data**. Each cruise sailing lasts for three days—departing at 7 pm on the first day and arriving at 8am on the last day, and only contacts during this period were studied. Embarking and disembarking begins and ends at approximately 12 noon on both days and devices are stored together prior to issuance or after collection. As such, data prior to departure and after arrival were not used, as the recorded data may be an artefact of devices being stored together.

All individuals onboard a cruise are issued a digital contact tracing device with a unique device identification number. These devices are calibrated based on signal strength to broadcast omnidirectional Bluetooth signals to other devices within a 2 m radius every 14.9 s followed by an omnidirectional scan of nearby signals lasting for 0.1 s. Each scan record captures the timestamp of the signal exchange and the identification number of the interacted device. After every five-minute interval, the records of 30 unique devices with the highest signal strength in each device are then stored. The stored records are then uploaded to a server on land. Further data processing is required to determine the duration of contact between two individuals. If there are two or more records with consecutive difference(s) of less than five minutes, the duration of the contact is the difference between the last and first timestamp in the series of records.

For each cruise sailing, we collected a de-identified manifest with the device identification number and details of the device holder (passenger or crew; for crew: department of the crew (Supplementary Table 2); for passengers: cabin number, keycard number, age, gender). The cruise ship can be demarcated into different areas based on the activities in a location (i.e. type of location: food and beverage (F&B), entertainment, shops, sports, public areas) and all passengers were required to tap-in using their keycards upon entering a new area onboard the cruise ship. We also collected a de-identified list of entry records with each record capturing the keycard number, location and timestamp of entry.

Using the three data sources (i.e. contact data, de-identified manifest and de-identified location records), we categorised the contacts between each dyad into one of four contact groups, $g$, namely (i) passenger-passenger contact from within the same travelling group (i.e. passengers in the same cabin or having a cumulative contact duration of more than 5 h over 3 days), (ii) passenger-passenger contact from different travelling groups, (iii) crew-crew contact, and (iv) passenger-crew contact. Five hours was selected as a conservative definition for travelling groups, given that this is considerably longer than an average meal duration and more than 99% of the cumulative contact duration (i.e. sum of all contact episodes) between passengers from different cabins were less than this duration.

We further classified a contact episode in a location into close, casual and transient types of contact if the cumulative duration of contact was at least 15 min, at least 15 min but less than 15 min, and less than 5 min respectively in a 2 m radius[39–41]. For each individual in each type of location, we estimated the number of different types of contacts (i.e. close, casual and transient contact) with passengers from different travelling groups over time spent in the location. Across the sailings, for each type of contact, we estimated the proportion of contacts occurring at a type of location over all types of location.

**Social network construction**. We performed a preliminary social network analysis and estimated the weighted degree distribution (number of contacts made per individual with each contact weighted by the duration of contact, to be elaborated), the distribution of the clustering coefficient (a measure of the triadic linkage among individuals[42]) and individuals' eigenvector centrality (a measure of direct and indirect centrality within a network) of passengers and crew in respective departments in each sailing. We performed a Welch's $t$-test to evaluate each network property for passengers against that for crew and $p$-values < 0.05 were

considered statistically significant. While the mean and interquartile range (IQR) of each estimate fluctuate across sailings, the 95% range of the estimates exhibit substantial overlap (Supplementary Fig. 1). Due to these similarities, we selected contact data collected over a single focal sailing with 1208 passengers and 1032 crew to construct the social network for simulating disease transmission for the primary analysis. However, we also carried out Supplementary analysis whereby simulations were also performed on all other sailings, and used this to ensure consistency in the percentage reduction in outbreak size for various outbreak interventions across the different sailings (Supplementary Fig. 7).

In the main analysis, we generated an undirected network with the strength of an edge weighted as a value between 0 and 1 based on the proportion of days with recorded contact over a three-day sail period and the exponent transformation of the mean daily cumulative contact duration between two individuals as follows:

$$w_{ij} = c_{ij}(1 - e^{-\bar{d}_{ij}\sigma}) \tag{1}$$

where $w_{ij}$ is the weight of a contact between individuals $i$ and $j$, $c_{ij}$ is the proportion of days with recorded contact and $\bar{d}_{ij}$ is the mean daily cumulative contact duration expressed hours. $\sigma$ is a scalar of 0.5 to approximate a scenario where the edge weight reaches 95% saturation after 3 h of contact ($w_{ij} \rightarrow 1$). As a sensitivity analysis, we explored other weightings for the network edges; similar to the above but 95% saturation to the same level of infection risk after 1 h of contact, or based on the proportion of days over the entire sailing with recorded contact only. These scenarios depict how risk of infection increases based on contact duration as observed in SARS-CoV-2 outbreaks in settings of poor ventilation[43,44] or transmission driven by a highly transmissible pathogen onboard cruises such as norovirus[45].

Incorporating $c_{ij}$ implicitly extends the contact networks as the contact data was collected over a 3-day sail but the transmission was simulated over a longer timescale of seven days to quantify the differences in outbreak trajectory for events lasting more than 3 days. Nevertheless, we have also performed sensitivity analysis using the actual temporal network to understand how the correlation of contact duration and sequence of contact events could potentially influence the outbreak.

**Transmission model**. We simulated SARS-CoV-2 Delta variant transmission on the above generated social contact network by extending the individual-based models developed by Firth et al. and Hellewell et al. (Table 3)[20,21].

For each simulation, we assume that the disease is introduced by one passenger who could be infected up to 14 days prior to the event, with equal probability on any of the days but the onset of the index case would only occur between the start (i.e. day 1) and the end (i.e. day 7) of the event. The distribution of the symptoms onset date, $S$, on respective day of the event, $d$, is as follows:

$$S(d) = \int_{-13}^{0} I(\delta)\theta(d + |\delta|)d\delta \tag{2}$$

where $\delta$ is the day of infection prior to the event (i.e. $\delta = 0$ represents the day before the start of event), $I(\delta)$ is the probability of being infected on any of the 14 days prior to the event and is fixed at 1/14, $\theta$ is the incubation period distribution with $d + |\delta|$ representing the time since infection on the respective day of the event.

Currently, all crew are required to be tested weekly and are largely confined to the cruise except during periods of shore leave, thereby reducing the risk of disease introduction by crew. Each day, the model searches for susceptible individuals in contact with the infected cases who are not isolated and infection from infector $i$ to susceptible individual $j$ occurs based on the following probability:

$$P_{i \rightarrow j}(d) = 1 - e^{-\Delta d \lambda_{i \rightarrow j}(d)} \tag{3}$$

where $\Delta d$ is the modelled time step of one day, and $\lambda_{i \rightarrow j}(d)$ is the force of infection between infector $i$ and susceptible individual $j$ on day $d$ expressed as:

$$\lambda_{i \rightarrow j}(d) = w_{ij}f(d|\mu_i, \alpha_i, \omega_i)r_{\text{scale}}\beta, \text{ for } \beta \in \{\beta^v, \beta^{mv}\} \tag{4}$$

where $f(d|\mu_i, \alpha_i, \omega_i)$ is the probability density function that represents the infectiousness of the infector on day $d$. We assumed a skew normal distribution with location parameter $\mu_i$ set based on the infector's day of onset, a slant parameter $\alpha_i$ and a scale parameter $\omega_i$ adjusted such that 25% of the infections occurred prior to symptom onset. As there is substantial uncertainty in the proportion of presymptomatic transmission for SARS-CoV-2[46], for sensitivity analysis, we considered a scenario where about 50% of transmission occurred prior to symptom onset. With a skewed normally distributed infectiousness profile centred based on the day of the symptoms onset, this ensures that the majority of the infections occurred around the time of symptoms onset[47,48].

While an edge weight has a maximum value of 1, infection between two individuals over the entire duration of infectiousness of the infected individual is not guaranteed. As such, we multiplied the force of infection with a scaling factor, $r_{\text{scale}}$, and this parameter was calibrated such that the mean probability of infection of a susceptible individual staying in the same cabin as an infected case is approximately 20% assuming exposure in the cabin and during all shared activities throughout the entire duration of infectiousness, similar to the household attack rates for SARS-CoV-2 Delta variant cases[34,49]. $\beta_{ij}$ is the relative risk of infection

depending on vaccination status and mask wearing behaviours, and is parameterised to reduce the probability of infection according to Table 4.

**Interventions**. In the testing interventions, the sensitivity of the tests were assumed to vary with viral load. We assumed PCR is 100% sensitive for cycle threshold (Ct) values (a measure of viral load) below 35 and rapid antigen tests are 94.5% sensitive for Ct values below 25 and lowered sensitivities as the Ct values increases[22]. The viral load trajectory was modelled in relation to the Delta variant, rising above the limits of test detection three days before symptoms onset with prolonged shedding post symptoms onset[12,23,24].

For the mask wearing intervention, the expected weight of the contact between individuals $i$ and $j$ of contact group $g$ are then modified based on the intervention of mask wearing and vaccination as follows:

$$\bar{\lambda}_{ijg}(t) = w_{ij} \int_{t-1}^{t} f(u;\mu_i,\alpha_i,\omega_i)du \, r_{scale}\left[(1-m_g)\beta^v + m_g\beta^{mv}\right] \quad (5)$$

where $m_g$ is the probability that the contact between any pairs of individual of a contact group $g$ occurs while wearing a mask. $\beta_v$ and $\beta_{m,v}$ are the relative risk of infection based on the vaccination status of the infector and infectee (Table 2).

For each intervention or combination of interventions, we ran 1000 simulations. We estimated the incidence by the day of infection, the number of cases in each generation, and the expected final outbreak size. All analyses were done in R version 4.0.4[50].

**Reporting summary**. Further information on research design is available in the Nature Research Reporting Summary linked to this article.

## Data availability
All data are available in the manuscript or the supplementary information. The data used for our analyses is publicly available at https://doi.org/10.5281/zenodo.6009027

## Code availability
The code used for our analyses is publicly available at https://doi.org/10.5281/zenodo.6009027

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

## Acknowledgements

R.P. acknowledges funding from the Singapore Ministry of Health. J.A.F. was supported by a research fellowship from Merton College and BBSRC (BB/S009752/1) and acknowledges funding from NERC (NE/S010335/1). A.J.K. was supported by a Sir Henry Dale Fellowship jointly funded by the Wellcome Trust and the Royal Society (grant 206250/Z/17/Z).

## Author contributions

Conceptualization: R.P., V.J.L. and A.J.K. Methodology: R.P., J.A.F., L.G.S. and A.J.K. Investigation: R.P. and Singapore CruiseSafe working group. Visualization: R.P., J.A.F., L.G.S. and A.J.K. Supervision: V.J.L. and A.J.K. Writing, original draft: R.P. and A.J.K. Writing, review & editing: all authors.

## Competing interests

The authors declare no competing interests.

## Additional information

## Singapore CruiseSafe working group

Annie Chang[8], Jade Kong[8], Jazzy Wong[8], Ooi Jo Jin[8], Deepa Selvaraj[1], Dominique Yong[1], Jocelyn Lang[1] & Abilash Sivalingam[9]

[8]Singapore Tourism Board, Singapore, Singapore. [9]Government Technology Agency, Singapore, Singapore.

## CMMID COVID-19 working group

Simon R. Procter[2], Stefan Flasche[2], William Waites[2], Kiesha Prem[2], Carl A. B. Pearson[2], Hamish P. Gibbs[2], Katharine Sherratt[2], C. Julian Villabona-Arenas[2], Kerry L. M. Wong[2], Yang Liu[2], Paul Mee[2], Lloyd A. C. Chapman[2], Katherine E. Atkins[2], Matthew Quaife[2], James D. Munday[2], Sebastian Funk[2], Rosalind M. Eggo[2], Stèphane Huè[2], Nicholas G. Davies[2], David Hodgson[2], Kaja Abbas[2], Ciara V. McCarthy[2], Joel Hellewell[2], Sam Abbott[2], Nikos I. Bosse[2], Oliver Brady[2], Rosanna C. Barnard[2], Mark Jit[2], Damien C. Tully[2],

Graham Medley[2], Fiona Yueqian Sun[2], Christopher I. Jarvis[2], Rachel Lowev, Kathleen O'Reilly[2], Sophie R. Meakin[2], Akira Endo[2], Frank G. Sandmann[2], W. John Edmunds[2], Mihaly Koltai[2], Emilie Finch[2], Amy Gimma[2], Alicia Rosello[2], Billy J. Quilty[2], Yalda Jafari[2], Gwenan M. Knight[2], Samuel Clifford[2] & Timothy W. Russell[2]

