## [Peer Review File · Nature Communications]

Using high-resolution contact networks to evaluate SARS-CoV-2 transmission and control in large-scale multi-day eventsREVIEWER COMMENTS

Reviewer #1 (Remarks to the Author):

This is an excellent empirical network analysis and modeling study to describe the contact patterns and SARS-CoV-2 transmission potential on cruise ships. It will likely have a very strong scientific impact, and the descriptive network data will be quite useful for other infectious disease modelers. A few comments or suggestions:

1. The empirical network data here is quite amazing, but more needs to be said about potential for missing data that could generate structural holes in the observed complete network (such holes would have a non-trivial impact on the network summary statistics like centrality). It is only mentioned briefly in the limitations. For example, what fraction of persons had no data, what there an attrition of use over time, was use of the sensors enforced?
2. The analysis of duration of contacts is helpful, but it does seem to assume that all contacts were "one-off", meaning that persons could come into contact for some minutes and then never contact each other for the duration of the sailing. How was the persistence of contacts handled, such as passengers in families who had the same contacts repeatedly with each other throughout the day? For example, one dyad could have a set of "transient" episodes throughout the three days that would accumulate.
3. Some of the results may not generalize to more realistic cruises due to the somewhat artificial nature of "cruises to nowhere" and the social distancing measures implemented in them. While there are several helpful sensitivity analyses related to the force of infection parameters, there were none on the network structure itself. The primary counterfactual scenario of interest to me would be what happens to the transmission dynamics and prevention interventions when the contact networks would scale to something more like a normal cruise? Although no contact data on that are available, a generative network model with scaling on the mean degree could approximate that well enough.
4. The mask risk reduction parameter (80%) is likely overestimated if passengers on the ship are wearing fabric masks and not expert-fitted N95 masks.
5. The discussion should/could include some mention of how the results are consistent or inconsistent with other transmission modeling studies of SARS-CoV-2 on cruise ships or related high-density environments.

Reviewer #2 (Remarks to the Author):

Summary article:

The article simulates the dynamics of a covid-19 outbreak (delta-variant) on a cruise of 7 days. The highlight of the model is the contact graph that is based on an analysis of contacts (using digital contact tracing devices) on four separated three-day sailings. The authors use literature data about transmissibility and effect of control measures (PCR test, antigen test, masks, and vaccination) to assess the effect of these control measures on an outbreak triggered by one primary infected case.

The central findings, as given in the discussion, are:

- structure and intensity of contacts have an impact on the effect of control measures
- contact structure if passengers on a cruise ship is different to e.g. a standard British community
- within 2-3 generations of infection, corona jumps from passengers to crew
- no combination of intervention measures did reduce the expected outbreak size below 1
- it is more or less possible to replace mask-wearing by antigen testing and a PCR test a day before the cruise

At the end, the authors critically discuss their study, and indicate particularly that the model does not take into account (1) contact tracing (2) completely asymptomatic cases (3) and that there are missing data about contact during water sports.

Overall comments:

In my opinion, the article has several strengths, as the detailed contact data, or the open discussion that clearly states the shortcomings of the model (I appreciate the open, scientific spirit of the authors in that point).

The paper seems to be very specific to a cruise, as the time horizon only is 7 days and the contact structure (as discussed also by the authors) is quite different from the general society. Therefore it is difficult to understand the implications of the results for policy-makers.

The model is based on a rich, detailed data set. However, the way the model is set up dismiss many of the detailed information. Particularly the auto-correlation in time between contacts is not taken into account, as infection are passed from one individual to another individual by a certain probability per day. The impact of that model structure only is touched in passing, though it is an important topic, necessary to understand the validity of the results. Particularly, as the sequence of contacts is known, it would have been possible to use the original contact data (perhaps repeating the days periodically or randomly), and in that, it would have not been necessary to relax to an abstract contact network. In my opinion, that's a missed chance.

As the detailed parameters of the infection and those of the control measures are not precisely known, a parameter uncertainty analysis should be included. There is some in the SI, but I would like to see a deeper analysis, and those results should also be presented in the main part of the paper.

Last comment: The authors do not mention data about outbreaks in the 3-days-test cruises. If infections have been discovered during these short test cruises it would be interesting to compare with the simulations. If there have been no infections detected, that would perhaps be worth to be mention.

Detailed comments:

I 38 there have been several attempts to reconstruct social networks form data (classical as well as digital contact data). Perhaps you want to link your study with those studies.

I 36/47 Bluetooth might have a problem to reliably measure the intensity of contacts, as the body itself shields the signal. If the body of one contactee is in-between the two devices, the contact might classified wrongly. Perhaps you want to comment on that.

I 78/90, Fig. 1C: If the contacts scale exponentially, the data in 1(c) should appear linearly. Does not really look like that (for my eyes). Moreover, perhaps you want to say something about the implication of that statement?

I 151-159 As indicated above, you have nice data, particularly incorporating information about autocorrelation and timing of contacts. If we consider a period of 7 days only, it might very well matter that during sleep there is only contact to the spouse, and also if the contact happens in the morning or evening. I'm a little unhappy with the aggregate model you generate from the nice data you have at hand. Perhaps you could discuss why this procedure is appropriate?

I 217-223 Discussion of the effect for different combinations of intervention measures. Perhaps you could summarize the effect by setting up a linear model (or a generalized linear model assuming e.g. a Poissonian distribution for the final size) that takes into account interactions of the measures. This could perhaps support the more informal verbal description.

I 318-322 Not only contact tracing is neglected, but also the response (reduction of contacts, self-isolation) of a person who knows that she has had a close contact to an infected person.

I 403-425 (a) Perhaps you can quote literature for your cluster coefficient (there are 1000 different version of cluster measures in the literature). (b) Implicitly, you claim that social networks which inherit the same aggregate macro-parameters (cluster coefficient etc.) behave similarly w.r.t. the outbreak of an epidemic. Perhaps you can support that claim?

I 421-424 You define your weight $w_{i,j}$ precisely, but you do not give a rational for this specific choice of the weight. There is some arbitrariness how to incorporate the number of contacts per day and the contact time. That should be based on a microscopic transmission model. Perhaps you can explain your choice?

I 435-507 This section (model description) is written from the perspective of a person who did develop the model, not from the perspective of a reader. Admittedly, only a minor fraction of readers will go through the details, but perhaps you could improve this section that all information is clearly stated and accessible (here I want to mention that I acknowledge that you make your code available, such that finally the interested reader can check by herself what you did in detail. Thanks for that!).

A few examples where I have the impression that the read could be improved: In maths papers it is considered as good style to first introduce a variable/parameter before using it. Here, the sequence often is turned around. In Tab 3 you mention the "scaling parameter" that later appears as r_{scale} without making the connection explicit. You do not define the function $\theta(.)$ used in the formula 449-451, nor do you indicate where it is defined (not the interpretation, that is

there, but the explicit shape). You use a skew normal distribution (l. 467-468) without giving a rationale (there is, e.g. the Gamma distribution used by other authors), perhaps you have a quotation for your reasons. The numerical values for the parameters of this function are not given. L 474-478: The scaling factor is applied to all contacts not only to close cabin contacts, why? And, a cabin will be more confined as a household, so I would naively expect that the transmission in a cabin is more likely? Notation $\beta_{i,j}$: You introduce multiple variants of beta (β_{vi} , β_{vj} ...) without using them subsequently. Seems to be unnecessary, as $\beta_{i,j}$ refers to two specific individuals i and j , and in that, depends on the properties of those individuals. It is sufficient to state the values of beta for the different cases. Same point, Table 4: Instead of stating the relative risk in numbers, you give them a parameter name (meaningless to the reader) and explain in the text (right column) the values. In line 493-395 suddenly a "contact group g " appears that is (to my knowledge) not defined before. Etc. All in all, I think it is worth to look at the appendix from the perspective of a reader and explain the model for her.

Summary:

The article is interesting in my opinion, but I have the impression that there are some technical issues that should be addressed. Furthermore, while the results might be interesting for organizers of cruises, I'm not sure what we can learn for the overall control of the delta variant in daily life. And, cruises are not known to be important factors for the spread of corona. All in all, I'm not sure if the article is suited for a journal as nature communications addressing a general audience; perhaps a more specialized journal would be better suited.

REVIEWER COMMENTS

Reviewer #1 (Remarks to the Author):

This is an excellent empirical network analysis and modeling study to describe the contact patterns and SARS-CoV-2 transmission potential on cruise ships. It will likely have a very strong scientific impact, and the descriptive network data will be quite useful for other infectious disease modelers. A few comments or suggestions:

1. The empirical network data here is quite amazing, but more needs to be said about potential for missing data that could generate structural holes in the observed complete network (such holes would have a non-trivial impact on the network summary statistics like centrality). It is only mentioned briefly in the limitations. For example, what fraction of persons had no data, what there an attrition of use over time, was use of the sensors enforced?

Thank you for these encouraging positive comments and for taking time to review this paper. The use of the contact tracing device (or sensors) among all passengers and crew were enforced by crew and officers from the Singapore Tourism Board (i.e. a third party independent of the cruise lines) to minimise attrition of use over time.

Loss of data could arise from functional issues of the device such as battery drainage and incomplete data upload but these could be minimised with proper training on device usage. Of the four sailings studied, the recovered data from sailings 1 and 2 covered more than 97% of all crew and passengers. As such, data from sailing 1 was used for the main disease transmission simulations to minimise the impact of missing data on the inference of the outbreak process.

Overall, data coverage of crew and passengers exceeded 90% in three sailings respectively. Given the heterogeneity in the data coverage across sailings, besides providing network summary statistics across all sailings (Table 2), we have illustrated the network summary statistics (including centrality) by respective sailings (Fig. S1 in original submission, attached in next page). Amendments to the discussion section are as follows:

Lastly, the accuracy of the data collected is largely dependent on the usage behaviour and the functionality of the device. Passengers are required to carry the contact tracing devices at all times except when engaged in water sports, and this was enforced by crew and external enforcement officers. Hence, interactions at the water sports areas may not be well represented but this effect to our analysis is expected to be minor as the cruise line of study required passengers to book these facilities in advance to facilitate crowd control. In a cabin, each passenger's device may not necessarily be placed in a 2m proximity and the frequent close contact interactions in these settings would not be recorded accurately. However, given that individuals in the cabin would largely continue to interact with each other outside the cabin while carrying the device, this would help to record a large proportion of their close interaction. ~~Thus~~ Furthermore, this limitation is reduced when the probability of infection saturates after a certain level of exposure as is in the case of SARS-CoV-2³⁸. Functional issues of the contact tracing device such as drainage of batteries and incomplete data uploading can affect the extent of missing data but could be minimised with proper training on device usage. In our main analysis, the chosen sailing had more than 97% coverage in both crew and passengers to minimise the impact of missing data on the inference of the outbreak dynamics. Outbreaks were also simulated in three other sailings as part of sensitivity analysis and similar trends in the outbreak trajectory was observed.

Fig. S1 Social network analysis over four cruise sailings. (A) Weighted degree, (B) eigenvector centrality, (C) clustering coefficient of crew and passengers of each sailing. Colours represent the cruise departure date and the median (shapes), 50% (dark lines) and 95% intervals (light lines) are shown. Weights were assigned based on exponent transformation of the mean daily cumulative duration of interaction between two individuals (see Materials and methods)

2. The analysis of duration of contacts is helpful, but it does seem to assume that all contacts were “one-off”, meaning that persons could come into contact for some minutes and then never contact each other for the duration of the sailing. How was the persistence of contacts

handled, such as passengers in families who had the same contacts repeatedly with each other throughout the day? For example, one dyad could have a set of “transient” episodes throughout the three days that would accumulate.

Thank you for raising this and we would like to clarify that we have quantified the weight of a contact between two individuals based on two aspects, (i) the mean daily cumulative contact duration, $\overline{d_{ij}}$, and (ii) the proportion of days with recorded contact over the three-day sail period, c_{ij} for $0 < c_{ij} < 1$.

For passengers i and j who are family members forming contacts repeatedly with each other throughout a day, we calculate the cumulative contact duration for that specific day, d_{ij} . We then used the mean cumulative contact duration on the days of contact, $\overline{d_{ij}}$, to construct the network.

For a dyad with a set of “transient” episodes throughout the three days, assuming a hypothetical scenario where fifteen 1-minute contact episodes were recorded each day over three days, d_{ij} for each day is 15 mins (hence $\overline{d_{ij}}$ is also 15 mins) and c_{ij} is 1 (because contact was made in three out of three days). By calculating the cumulative contact duration for a day, we assumed that the risk of acquiring an infection is a cumulative process (i.e. risk does not reset to zero when there is a break in the contact) and the risk of transmitting an infection does not change across the hours of a day. As such, multiple transient contacts can be as risky as a single close contact. This assumption was made as there is a lack of epidemiological studies to suggest lowered risk of SARS-CoV-2 transmission over intermittent contacts episodes as opposed to continuous contact. Operationally, we have also defined a close contact as an individual with cumulative exposure of at least 15 mins to an infected individual, thus influencing our model design.

Taking into consideration comments from both reviewers, we have added another sensitivity analysis on the outbreak dynamics on a temporal network. The model searches through the edge list to identify susceptible contacts in a day to simulate infection based on the contact duration and the infectiousness of the infected individual in that specific day. Our new results showed a lowered outbreak size as the simulation lasted for 3 days only. However, the general trend in the expected outbreak size over increasing vaccine coverage and the reduction in outbreak size between a mask-on and mask-off remains unchanged.

As SAR-CoV-2 infected individuals have an incubation period, they are not likely to infect others on the same day of infection. Simulating outbreaks on a temporal network would account for the effect of heterogeneous daily cumulative contact duration on the outbreak size but the heterogeneous contact duration of contact episodes in a day would have negligible effect (i.e. probability of transmission over three contact episodes lasting 3, 5, 7-mins in a day would be the same as a single 15-min contact episode in a day). The median final outbreak size remains similar to that of a static network but a lower 95th percentile was observed for the outbreaks simulated on a temporal network (Fig S9 and S10). As passengers engaged in more activities on day 2 on the cruise sailing as compared to day 1, the number of contacts and duration is correspondingly higher. A static network that averages out these heterogeneity in contact could thus allow for a higher potential of transmission in earlier stages of the cruise sailing, resulting in a higher 95th percentile as compared to a temporal network (Fig S9 and S10). Nevertheless, it is encouraging to note that the median outbreak size is similar for outbreaks simulated in both a static and temporal network (Fig S9 and S10) as simulations on longer time scales of 7 days were performed in the main analysis using the static network which served as a means of extending the network beyond 3 days.

Fig. S9 Average and 95th percentile in outbreak size for varying interventions and vaccination coverage for outbreaks simulated on the temporal network for 3 days of sailing. Vaccines were assumed to confer 50% protection against infection and 50% lowered infectiousness for breakthrough infections in vaccinated individuals. Presymptomatic transmission was modelled to occur in 25% of the infections. (A, D) Daily edge weights vary based on the duration of contact with weights increasing with contact time but reaches 95% saturation after 3 hours of contact, (B, E) same as (A, D) but reaches 95% saturation after 1 hour of contact, (C, F) edge weights of 1 when interaction is recorded.

Fig. S10 Average and 95th percentile in outbreak size for varying interventions and vaccination coverage for outbreaks simulated on the static network for 3 days of sailing. Vaccines were assumed to confer 50% protection against infection and 50% lowered infectiousness for breakthrough infections in vaccinated individuals. Presymptomatic transmission was modelled to occur in 25% of the infections. (A, D) Daily edge weights vary based on the duration of contact with weights increasing with contact time but reaches 95% saturation after 3 hours of contact, (B, E) same as (A, D) but reaches 95% saturation after 1 hour of contact, (C, F) edge weights of 1 when interaction is recorded.

3. Some of the results may not generalize to more realistic cruises due to the somewhat artificial nature of “cruises to nowhere” and the social distancing measures implemented in them. While there are several helpful sensitivity analyses related to the force of infection parameters, there were none on the network structure itself. The primary counterfactual scenario of interest to me would be what happens to the transmission dynamics and prevention interventions when the contact networks would scale to something more like a normal cruise? Although no contact data on that are available, a generative network model with scaling on the mean degree could approximate that well enough.

Thank you very much for raising this. While many previous studies have focussed on simulated networks. A major aspect and focus of this particular work is the use of empirical networks to analyse disease transmission under different outbreak interventions. We do agree that it is possible to scale up the number of crew and passengers onboard, the degree of contacts and to assign social links to resemble a normal cruise setting. However, this implies a switch to (or increased focus on) a simulation based network analysis, and we currently hope to keep the focus on empirical networks. Nevertheless, in terms of future work, and given the increased use of Bluetooth devices in measuring contacts at large scale events, we are planning to perform a subsequent deeper comparison of contact networks in different interaction settings, and how different network parameters influence an outbreak under different transmission parameters. But, this will be carried out as a separate, and extensive, study in the future.

We realise it would be helpful to conduct additional sensitivity analysis on the network structure by scaling the size of the network. Nevertheless, we would like to share that different ways of assigning the weights of an edge (range 0 to 1) were performed in the current study with weights in a day approaching 1 after 3 hours of contact and equal to 1 upon any contact corresponding to the most and least conservative assumptions. Conceptually, individual differences in their degree of contact is thus included in the analysis although one other factor that is not tested (until further studies are performed) is a potential change in the clustering of contacts under a typical cruise or other large scale events.

Taking into consideration the importance of comparing the current network structure with other events, we would like to propose the following edits to the limitations section:

Despite such limitations, this is one of the few studies with large- and fine-scale data collection from multiple events in one setting and comparison with future studies of similar data collection methods in ~~these~~ similar and other settings may help to strengthen our findings and provide better understanding of transmission dynamics under different network structure and disease characteristics.

4. The mask risk reduction parameter (80%) is likely overestimated if passengers on the ship are wearing fabric masks and not expert-fitted N95 masks.

All passengers were required to wear a surgical mask that was distributed by the cruise line when boarding the cruise. In the same systematic review and meta-analysis by Chu *et al.*, an N95 respirator was approximately 96% effective in protecting the wearer. Surgical mask was about 67% effective in protecting the wearer and about 70% effective in a non-healthcare setting. Keeping all other parameters unchanged and reducing the protective effect of wearing a mask from 80% to 70%, the following figure illustrates the outbreak trajectory. The key difference from Fig 4 in main text or Fig S4 (for outbreaks assuming 80% mask risk reduction) is the outbreak size but the general trajectory across interventions remains unchanged.

Based on the comments from both reviewers, we have incorporated a further sensitivity analysis by simultaneously varying parameters related to the infection process and control measures. Please refer to replies to reviewer #2 below (Overall comments #3) for more details.

Average and 95th percentile in outbreak size for varying interventions, vaccination coverage and assumption on network edge. Vaccines were assumed to confer 50% protection against infection and 50% lowered infectiousness for breakthrough infections in vaccinated individuals. Mask was assumed to be 70% effective in preventing transmission, when both the infector and infectee are wearing a mask. Presymptomatic transmission was modelled to occur in 25% of the infections. (A, D) Edge weights vary based on the proportion of days with recorded interaction over a three-day sail period and duration of contact with weights increasing with days of interaction and contact time but reaches 95% saturation after 3 hours of contact, (B, E) same as (A, D) but reaches 95% saturation after 1 hour of contact, (C, F) edge weights vary based on proportion of days with recorded interaction.

5. The discussion should/could include some mention of how the results are consistent or inconsistent with other transmission modeling studies of SARS-CoV-2 on cruise ships or related high-density environments.

Thank you for raising this. We have edited parts of the discussion to compare with other observational and modelling studies.

With numerous work functions interfacing with passengers, and given the overlapping shifts and closely related job scope between crew (e.g. F&B and galley, hotel services and housekeeping), we found it only took around 32 generations for the infection to spread from a passenger to a crew and an additional generation of transmission to before reaching another crew in a different department. For SARS-CoV-2 transmission on the Diamond Princess cruise ship, the earliest onset in crew occurred about 18 days after the onset of the index case²⁰. Assuming a generation time of about 5–7 days, this corresponds to a spillover from passengers to crew after three to four generations of transmission. With about 2.6 times more passengers than crew on the Diamond Princess cruise ship, this could delay the spillover in disease transmission. Crew and event personnel play an important role in ensuring smooth operations and their wellbeing should be accounted for in the plans when

reopening events. Hence, besides encouraging crew cohorting, interventions that minimise transmission in passengers would have an indirect effect of protecting the crew.

...

The expected outbreak size under different combinations of interventions was sensitive to the assumptions of the network edge weights. When edges are weighted by the proportion of days with recorded interaction over a three-day sail period, two individuals with transient contact in a day are assumed to have the same risk as two individuals with close contact in a day. This assumption is applicable when the dominant mode of transmission is largely independent of the duration of contact (e.g. environmental or airborne transmission). There were 5 times more transient interactions than close contacts and these contacts are now equally at risk of infection. Thus, mask wearing would largely help to lower the risk of transmission and acquiring infection, and outperforms PCR testing or even twice antigen testing interventions especially when the proportion of presymptomatic transmission is high during the early stage of viral shedding. Overall, we estimate that mask wearing with passengers practising physical distancing could reduce transmission by about 54%, approximately 20–30% lower than the effectiveness of wearing surgical mask in healthcare settings or in public areas after accounting for interactions in mask-off settings when dining or engaged in sports. This findings corroborates with behavioural surveys reporting 1.3-2 times higher risk of being infected when mask wearing in enclosed spaces is not practised²⁴. Our risk reduction estimates are about 5 times higher than that reported in a cluster-randomised trial. However, in this trial proper mask wearing occurred in less than half of that intervention arm, thereby limiting the multiplicative effectiveness of mask wearing in reducing both infection and transmission²⁵.

Both mask-off, rapid antigen testing and mask-on, once-off PCR testing would help to reduce the risk of disease introduction and further transmission if the index cases successfully escape initial detection. While their differences in the expected outbreak size was less than 10 cases across different assumptions to the edge weights, they bring different outcomes to the passenger experience and operations planning — an extra swab test at the middle of the event versus wearing a mask at all times other than during dining and engaged in sports, logistics to check the test outcomes versus monitoring mask wearing practisespractices, managing false positives versus passengers flouting rules. Pre-event rapid antigen testing has been widely adopted in many large-scale events lasting less than a day and accounted for about 53% reduction in transmission in settings with high levels of social contacts and about 72% reduction after accounting for physical distancing¹. In a fully susceptible cohort, we estimate a mask-off, rapid antigen testing intervention at the start and midway of the event would reduce the mean outbreak size by over 90% with the additional reduction largely attributed to the administration of an additional test.

Reviewer #2 (Remarks to the Author):

Summary article:

The article simulates the dynamics of a covid-19 outbreak (delta-variant) on a cruise of 7 days. The highlight of the model is the contact graph that is based on an analysis of contacts (using digital contact tracing devices) on four separated three-day sailings. The authors use literature data about transmissibility and effect of control measures (PCR test, antigen test, masks, and vaccination) to assess the effect of these control measures on an outbreak triggered by one primary infected case.

The central findings, as given in the discussion, are:

- structure and intensity of contacts have an impact on the effect of control measures
- contact structure if passengers on a cruise ship is different to e.g. a standard British community
- within 2-3 generations of infection, corona jumps from passengers to crew
- no combination of intervention measures did reduce the expected outbreak size below 1
- it is more or less possible to replace mask-wearing by antigen testing and a PCR test a day before the cruise

At the end, the authors critically discuss their study, and indicate particularly that the model does not take into account (1) contact tracing (2) completely asymptomatic cases (3) and that there are missing data about contact during water sports.

Overall comments:

1. In my opinion, the article has several strengths, as the detailed contact data, or the open discussion that clearly states the shortcomings of the model (I appreciate the open, scientific spirit of the authors in that point). The paper seems to be very specific to a cruise, as the time horizon only is 7 days and the contact structure (as discussed also by the authors) is quite different from the general society. Therefore it is difficult to understand the implications of the results for policy-makers.

Thank you for agreeing to review this paper. We have revised our analysis and provided clarification to the comments. For ease of tracking, we have enumerated the comments.

While the study was based on cruise contact networks, the myriad of activities on board and various groups of crew and passengers is similar to other large-scale events (e.g. conferences). The study was an attempt to understand how contact networks and interventions interact to influence outbreak patterns but we agree that more data collected under other settings is necessary for comparison. We have made revisions to our limitation sections to reflect this and an extract of the edits could be found in the reply to Review #1's third comment.

2. The model is based on a rich, detailed data set. However, the way the model is set up dismiss many of the detailed information. Particularly the auto-correlation in time between contacts is not taken into account, as infection are passed from one individual to another individual by a certain probability per day. The impact of that model structure only is touched in passing, though it is an important topic, necessary to understand the validity of the results. Particularly, as the sequence of contacts is known, it would have been possible to use the original contact data (perhaps repeating the days periodically or randomly), and in that, it would have not been necessary to relax to an abstract contact network. In my opinion, that's a missed chance.

Taking into consideration comments from both reviewers, we have added another sensitivity analysis on the outbreak dynamics on a temporal network. The model searches through the edge list to identify susceptible contacts in a day to simulate infection based on the contact duration and the infectiousness of the infected individual in that specific day. Our new results showed a lowered outbreak size as the simulation lasted for 3 days only. However, the general trend in the expected outbreak size over increasing vaccine coverage and the reduction in outbreak size between a mask-on and mask-off remains unchanged.

We agree that from the data, insights on the sequence of contacts over the days and within a day can be derived. As SAR-CoV-2 infected individuals have an incubation period, they are not likely to

infect others on the same day of infection. Simulating outbreaks on a temporal network would account for the effect of heterogeneous daily cumulative contact duration on the outbreak size but the heterogenous contact duration of contact episodes in a day would have negligible effect (i.e. probability of transmission over three contact episodes lasting 3, 5, 7-mins in a day would be the same as a single 15-min contact episode in a day) and the sequence of these contact episodes in a day would have negligible effect on subsequent transmission which is expect to happen days after initial infection.

The median final outbreak size remains similar to that of a static network but a lower 95th percentile was observed for the outbreaks simulated on a temporal network (Fig S9 and S10). As passengers engaged in more activities on day 2 on the cruise sailing as compared to day 1, the number of contacts and duration of contact are correspondingly higher. A static network that averages out these heterogeneity in contact could thus allow for a higher potential of transmission in earlier stages of the cruise sailing, resulting in a higher 95th percentile as compared to a temporal network (Fig S9 and S10). Nevertheless, it is encouraging to note that the median outbreak size is similar for outbreaks simulated in both a static and temporal network (Fig S9 and S10) as simulations on longer time scales of 7 days were performed using the static network which served as a means of extending the network beyond 3 days. We have added these outcomes in the results sections.

Fig. S9 Average and 95th percentile in outbreak size for varying interventions and vaccination coverage for outbreaks simulated on the temporal network for 3 days of sailing. Vaccines were assumed to confer 50% protection against infection and 50% lowered infectiousness for breakthrough infections in vaccinated individuals. Presymptomatic transmission was modelled to occur in 25% of the infections. (A, D) Daily edge weights vary based on the duration of contact with weights increasing with contact time but reaches 95% saturation after 3 hours of contact, (B, E) same as (A, D) but reaches 95% saturation after 1 hour of contact, (C, F) edge weights of 1 when interaction is recorded.

Fig. S10 Average and 95th percentile in outbreak size for varying interventions and vaccination coverage for outbreaks simulated on the static network for 3 days of sailing. Vaccines were assumed to confer 50% protection against infection and 50% lowered infectiousness for breakthrough infections in vaccinated individuals. Presymptomatic transmission was modelled to occur in 25% of the infections. (A, D) Daily edge weights vary based on the duration of contact with weights increasing with contact time but reaches 95% saturation after 3 hours of contact, (B, E) same as (A, D) but reaches 95% saturation after 1 hour of contact, (C, F) edge weights of 1 when interaction is recorded.

3. As the detailed parameters of the infection and those of the control measures are not precisely known, a parameter uncertainty analysis should be included. There is some in the SI, but I would like to see a deeper analysis, and those results should also be presented in the main part of the paper.

In most of the sensitivity analysis, we aim to understand how a certain aspect of the infection process and the outbreak intervention influence the final outbreak size. We agree that in reality, uncertainty to the parameters of the infection process and the interventions occur in tandem. We have performed a sensitivity analysis with the following assumptions:

Table S1.

Parameter uncertainty for Fig S8, assuming uniform distribution across the assumed values

Parameter	Assumed values	Details and references
Pre-symptomatic transmission	25-50%	¹

Adherence to isolation when tested positive	60-100%	For scenarios involving testing only, we assume that there are available cabins for individuals to isolate given that cruises are operating at 50% capacity. Lower bound based on self-reported adherence to isolation in the UK ² .
Relative risk of transmission by mask-off vaccinated, infected individual i	50-100%	Mean probability of transmitting infection reduces by 0-50% ^{3,4} .
Relative risk of acquiring infection by mask-off vaccinated, susceptible individual j	30-50%	Mean probability of acquiring infection reduces by 50-70% ⁴⁻⁹ .
Relative risk of transmission when both infected individual i and susceptible individual j are wearing a mask	20-60%	Mean probability of infection reduces by about 40-80% when both the infected individual and susceptible contact are wearing a mask ¹⁰ .

Our results showed that the expected and 95th percentile of the outbreak size of each scenario falls between that observed in simulations with 25% transmission occurring prior to symptoms and in simulations with 50% transmission occurring prior symptoms. While the risk reduction through a mask-on intervention now has a wider uncertainty of 40-80%, we have also introduced a wider uncertainty in the adherence to isolation (60-100%) after testing positive. As such, the outbreak size for a mask-on, no testing scenario and a mask-off, one-off PCR testing scenario are both worse off than previous assumptions and their differences are also narrowed. The lowered adherence to isolation coupled with the possibility of vaccinated infected individuals being as infectious as unvaccinated individuals resulted in larger outbreak size observed in all testing interventions at low vaccine coverage than at high vaccine coverage. This was in spite of the potential of vaccinated susceptible contacts having a higher risk reduction against infection of 50-70% which counteracts the impacts of the aforementioned interventions.

Fig S8. Average and 95th percentile in outbreak size for varying interventions, vaccination coverage, assumption on network edge and variation of transmission parameters and effectiveness of control measures based on Table S2. (A, D) Edge weights vary based on the proportion of days with recorded interaction over a three-day sail period and duration of contact with weights increasing with days of interaction and contact time but reaches 95% saturation after 3 hours of contact, (B, E) same as (A, D) but reaches 95% saturation after 1 hour of contact, (C, F) edge weights vary based on proportion of days with recorded interaction.

4. Last comment: The authors do not mention data about outbreaks in the 3-days-test cruises. If infections have been discovered during these short test cruises it would be interesting to compare with the simulations. If there have been no infections detected, that would perhaps be worth to be mention.

Thanks for this final comment. There was no outbreak in the four sailings studied and while a COVID-19 case was detected in one of the sailings since the resumption of cruise operations, there was no further transmission. An additional elaboration of this has been added into the results section, worded as follows:

Based on real life cruise operations, with over 80% of the population receiving two doses of COVID-19 vaccination and the implementation of a one-off pre-event rapid antigen testing, no outbreaks occurred on cruises even when the reported community incidence was 0.7 per 1,000 at the height of the outbreak in end October 2021 which was approximately 30% lower than that simulated in the model (i.e. one initial infected passenger corresponding to a community incidence of about 1 per 1,000).

Detailed comments:

5. I 38 there have been several attempts to reconstruct social networks form data (classical as well as digital contact data). Perhaps you want to link your study with those studies.

New citations, including a systematic review, relating to those studies were added as follows:

However, the transmission dynamics on real world networks of large-scale events are yet to be fully explored in the COVID-19 era⁶. Furthermore, while pre-COVID-19 studies on human contact networks for the transmission of close contact infections have analysed various network properties and attempted to reconstruct the social network from contact diaries or digital sensors, they are largely focused in school, healthcare settings or the greater community with few studies on conferences and business meetings⁷⁻¹⁰.

6. I 36/47 Bluetooth might have a problem to reliably measure the intensity of contacts, as the body itself shields the signal. If the body of one contactee is in-between the two devices, the contact might be classified wrongly. Perhaps you want to comment on that.

Currently, the broadcast and scanning of Bluetooth signals in these devices are omnidirectional. If the body of one contactee (individual C) is between two devices of two other individuals (individuals A and B) as illustrated below, the device in A is still capable of receiving signals from B so long as B is within a 2m radius. The strength of B's signal captured by A will be weaker than the strength of C's signal which is representative of the proximity of contact but, regardless, both are considered as contacts of A.

We have made minor edits this in the material and methods section as follows:

These devices are calibrated based on signal strength to broadcast omnidirectional Bluetooth signals to other devices within a 2m radius every 14.9 seconds followed by an omnidirectional scan of nearby signals lasting for 0.1 seconds.

7. I 78/90, Fig. 1C: If the contacts scale exponentially, the data in 1(c) should appear linearly. Does not really look like that (for my eyes). Moreover, perhaps you want to say something about the implication of that statement?

Extracting the median number of close contacts for various threshold and applying the log transformation, we obtain the values in the table below:

Threshold (mins)	5	10	15	20	25	30
Median close contacts	39.6	22.1	14.3	10.4	8.4	6.5
Log median close contacts	3.7	3.1	2.7	2.3	2.1	1.9
Factor of increase with every 5 min reduction in threshold for close contact	1.2	1.1	1.2	1.1	1.1	-

The factor of increase in the log median close contact is largely constant for every 5-min decrease in the threshold for close contact definition, thus the contacts appear to scale exponentially.

The changes in the threshold for close contact would have implications on contact tracing and we have raised this implication under the discussion section (underlined).

One limitation to our study was that we did not model contact tracing around detected cases and the behaviour of contacts who are aware of their potential exposures. While cruise lines are trained to trace and quarantine close contacts as part of the pilot reopening, as the ease of rapid testing increases with fast turnaround time, this could serve as a replacement for slower and resource intensive contact tracing in such settings. With pre-symptomatic transmission of SARS-CoV-2 and high levels of transmissibility of the Delta variant, the effectiveness of contact tracing is approaching a point of saturation in many countries²⁶. Furthermore, even if the threshold for close contact to be traced is lowered, the corresponding exponential increase in contacts fulfilling this criteria would make it logistically challenging to trace all individuals in a reasonable amount of time.

8. | 151-159 As indicated above, you have nice data, particularly incorporating information about autocorrelation and timing of contacts. If we consider a period of 7 days only, it might very well matter that during sleep there is only contact to the spouse, and also if the contact happens in the morning or evening. I'm a little unhappy with the aggregate model you generate from the nice data you have at hand. Perhaps you could discuss why this procedure is appropriate?

Thank you for raising this and the results from sensitivity analysis using the temporal contact network is as mentioned above.

Secondary, and some tertiary, transmission is observed when simulating transmission using the actual 3-day contact network. However, the outbreak size was small for meaningful comparison of various outbreak interventions. When extending the timescale of the outbreak simulation, a corresponding extension of the contact network is necessary. Thus, we decided to generate an aggregated contact network model instead of generating a hypothetical network by sampling the contacts and duration of contacts for each individual on days with no data collected.

However, taking into consideration an earlier comment #2, we have performed additional analysis using the actual temporal network and have elaborated our findings in the reply above.

9. | 217-223 Discussion of the effect for different combinations of intervention measures. Perhaps you could summarize the effect by setting up a linear model (or a generalized linear model assuming e.g. a Poissonian distribution for the final size) that takes into account interactions of the measures. This could perhaps support the more informal verbal description.

While the combination of intervention measures are independent, the final size of the outbreaks were dependent on the chains of transmission and hence the network structure. In other words, for the same testing and mask wearing intervention, the sample of final sizes across different vaccine coverage would need to be controlled for the impact of clustering and degree of contacts. A linear (or generalised linear model) may not be sufficient to tease out the impact to the final outbreak size arising from the network structure or the outbreak interventions. Furthermore, the different combinations of interventions may interact in a nonlinear manner. Thus for clarity, we retain the original plots and text instead of introducing a linear model.

10. | 318-322 Not only contact tracing is neglected, but also the response (reduction of contacts, self-isolation) of a person who knows that she has had a close contact to an infected person.

Thank you for raising this. We agree that an individual's awareness of having a close contact with an infected person may prompt the individual to reduce their number of contacts or undergo non-mandatory self-quarantine. Data on adherence to self-isolation for infected individuals are available (and new sensitivity analysis was added) but much less is known about self-quarantine for close contacts – especially if this quarantine is non-mandatory. While we did not elaborate this in detail in the discussion, we have added following:

One limitation to our study was that we did not model contact tracing around detected cases and the behaviour of contacts who are aware of their potential exposures. Thus, our estimates serve as an upper bound to the potential outbreak size.

11. | 403-425 (a) Perhaps you can quote literature for your cluster coefficient (there are 1000 different version of cluster measures in the literature).

Thank you for mentioning this. We have a citation for (a) as follows:

We performed a preliminary social network analysis and estimated the weighted degree distribution (number of contacts made per individual with each contact weighted by the duration of contact, to be elaborated), the distribution of the clustering coefficient (a measure of the triadic linkage among individuals³³)

(b) Implicitly, you claim that social networks which inherit the same aggregate macro-parameters (cluster coefficient etc.) behave similarly w.r.t. the outbreak of an epidemic. Perhaps you can support that claim?

Firstly, we apologise if the previous wording of the text implied this point highlighted by the reviewer here. This was not our intention and we have reworded this to ensure this part of the text is clearer. We would also like to clarify that our intention here was to identify a few key characteristics of the network for presentation. Other factors such as the contact matrices between crew and passengers, higher order clustering and other network measures such as betweenness centrality were not reflected in this study but obviously could potentially influence

the outbreak dynamics for different networks. Thus, while our main analysis was based on the network from one cruise sailing, we have also performed supplementary simulations on the contact networks of three other cruise sailings to ensure consistency in the results (See Fig S7). We also reference this supplementary analysis in the main text in the appropriate places.

12. I 421-424 You define your weight $w_{i,j}$ precisely, but you do not give a rationale for this specific choice of the weight. There is some arbitrariness how to incorporate the number of contacts per day and the contact time. That should be based on a microscopic transmission model. Perhaps you can explain your choice?

The two measures of contact which we could derive from the data was the contact time and the frequency of contact over the 3-day sails. In general, as contact time increases, the weight or strength of a contact will increase. However, there is uncertainty in what constitutes an effective contact for SARS-CoV-2 transmission. It is influenced by the duration of contact, and the strength of this contact could be quantified in binary terms (0 or 1 only) or over a continuum (0 to 1) with varying rate of increase in strength over time. Due to the arbitrariness, we decided to model transmission across different weighted networks with contact weight (i) saturating after 3 hours, (ii) saturating after 1 hours and (iii) effectively established once contacted as mentioned in the methods section.

Furthermore, as we attempt to model transmission over a 7-day period using a 3-day contact network, we implicitly extend the network by factoring in the proportion of days where contact was recorded and have elaborated in the manuscript as follows:

where w_{ij} is the weight of a contact between individuals i and j , c_{ij} is the proportion of days with recorded contact and $\overline{d_{ij}}$, is the mean daily cumulative contact duration expressed hours. σ is a scalar of 0.5 to approximate a scenario where the edge weight reaches 95% saturation after 3 hours of contact ($w_{ij} \rightarrow 1$). As a sensitivity analysis, we explored other weightings for the network edges; similar to the above but 95% saturation to the same level of infection risk after 1 hour of contact, or based on the proportion of days over the entire sailing with recorded contact only. These scenarios depict how risk of infection increases based on contact duration as observed in SARS-CoV-2 outbreaks in settings of poor ventilation^{34,35} or transmission driven by a highly transmissible pathogen onboard cruises such as norovirus³⁶.

Incorporating c_{ij} implicitly extends the contact networks as the contact data was collected over a 3-day sail but the transmission was simulated over a longer timescale of seven days to quantify the differences in outbreak trajectory for events lasting more than 3 days. Nevertheless, we have also performed sensitivity analysis using the actual temporal network to understand how the correlation of contact duration and sequence of contact events could potentially influence the outbreak.

13. I 435-507 This section (model description) is written from the perspective of a person who did develop the model, not from the perspective of a reader. Admittedly, only a minor fraction of readers will go through the details, but perhaps you could improve this section that all information is clearly stated and accessible (here I want to mention that I acknowledge that you

make your code available, such that finally the interested reader can check by herself what you did in detail. Thanks for that!).

A few examples where I have the impression that the read could be improved: In maths papers it is considered as good style to first introduce a variable/parameter before using it. Here, the sequence often is turned around. In Tab 3 you mention the “scaling parameter” that later appears as r_{scale} without making the connection explicit. You do not define the function $\theta(.)$ used in the formula 449-451, nor do you indicate where it is defined (not the interpretation, that is there, but the explicit shape).

Apologies for the oversight. We have since added the notation r_{scale} and mentioned that a log normal distribution was used to model the incubation period in Table 3.

14. You use a skew normal distribution (l. 467-468) without giving a rational (there is, e.g. the Gamma distribution used by other authors), perhaps you have a quotation for your reasons. The numerical values for the parameters of this function are not given.

For SARS-CoV-2, a study by Ferretti *et al* has shown strong positive correlation between the generation interval (time of infection in index case to time of infection in secondary case) and the incubation period of the index case. In other words, individuals with a faster increase in pathogen load are likely to start transmitting earlier and also have a shorter incubation period.

A gamma, lognormal or Weibull distribution supports positive values and can be used to model infectiousness by the time of infection. Furthermore, the incubation period was modelled as an independent distribution in our study. However, without a joint distribution to account for the correlation of timing of infections and the incubation period, we run the risk of simulating infected individuals with their peak in infectiousness occurring many days ahead of their onset of symptoms or vice versa.

Our skewed normal distribution of the infectiousness was designed to ensure that the majority of the infections occurred around the time of the symptom onset and to allow for the possibility of pre-symptomatic infections. The values of slant parameter α_i and the scale parameter ω_i were both adjusted to allow for 25% or 50% of the transmission to occur prior to symptoms onset. These values are available in the code but were not mentioned in the manuscript as their conceptual meaning should be interpreted as in the previous sentence. We have elaborated in the manuscript as follows:

where $f(d | \mu_i, \alpha_i, \omega_i)$ is the probability density function that represents the infectiousness of the infector on day d . We assumed a skew normal distribution with location parameter μ_i set based on the infector’s day of onset, a slant parameter α_i and a scale parameter ω_i adjusted such that 25% of the infections occurred prior to symptom onset. As there is substantial uncertainty in the proportion of presymptomatic transmission for SARS-CoV-2⁴², for sensitivity analysis, we considered a scenario where about 50% of transmission occurred prior to symptom onset. With a skewed normally distributed infectiousness profile centred based on the day of the symptoms onset, this ensures that the majority of the infections occurred around the time of symptoms onset^{43,44}.

15. L 474-478: The scaling factor is applied to all contacts not only to close cabin contacts, why? And, a cabin will be more confined as a household, so I would naively expect that the transmission in a cabin is more likely?

Apologies that we did not go into detail on the rationale of the scaling factor, how it's parameterised and why it applies to all contacts. In most disease transmission network models, a scaling parameter ensures that the initial transmission mimics community observations to reproduce a similar basic reproduction number in a fully susceptible population. Such a parameter is applied to all individuals in the population and was used in an earlier simulation study by Firth *et al.* as cited in our paper.

In a cruise or large-scale event setting, the mean degree of contact was observed to be higher and hence the basic reproduction number, R_0 , is expected to be larger than that observed in the community but the exact value is unknown. Instead, we have information on the type of contacts (contacts formed between passengers of the same cabin, between passengers of different cabins, between crew etc). As such, using the edge weights and a different scaling factor, we could tune the probability of infection between two individuals in the same cabin over the entire duration of infectiousness to be approximately equal to the household attack rate. This scaling factor is necessary as the edge weights vary from 0 to 1 (typically close to 1 for same cabin contacts given prolonged interaction) but the probability of infection (or in this case the household attack rate) is only about 0.2 (or 20%).

Having tuned the probability of infection between two passengers in the same cabin, the same scaling factor is used to derive the force of infection between two passengers of different cabins, which is expected to be low given that the majority of the interactions lasted less than 5 minutes.

We agree that a cabin is more confined than a household which could increase the risk of transmission per unit of contact time. However, passengers on a cruise are also engaged in a variety of activities outside of the cabin (accompanied by passengers of the same cabin, of different cabins, or alone) over long periods. In the absence of granular cruise outbreak data, we are unable to definitively conclude that out-of-cabin contacts are less risky than in-cabin contacts.

We have edited the manuscript slightly as follows to explain why the scaling parameter is applied to all contacts:

While an edge weight has a maximum value of 1, infection between two individuals over the entire duration of infectiousness of the infected individual is not guaranteed. As such, we multiplied the force of infection with a scaling factor, r_{scale} , and this parameter was calibrated such that the mean probability of infection of a susceptible individual staying in the same cabin as an infected case is approximately 20% assuming exposure in the cabin and during all shared activities throughout the entire duration of infectiousness, similar to the household attack rates for SARS-CoV-2 Delta variant cases^{23,41}.

16. Notation $\beta_{i,j}$: You introduce multiple variants of beta (β_{vi} , β_{vj} ...) without using them subsequently. Seems to be unnecessary, as $\beta_{i,j}$ refers to two specific individuals i and j , and in that, depends on the properties of those individuals. It is sufficient to state the values of beta for the different cases.

Same point, Table 4: Instead of stating the relative risk in numbers, you give them a parameter name (meaningless to the reader) and explain in the text (right column) the values.

Thanks for this comment and we have amended Table 4 to reduce the number of variants of beta.

17. In line 493-395 suddenly a “contact group g” appears that is (to my knowledge) not defined before. Etc.

Thank you for raising this. The contact groups were mentioned earlier in the Data sub section of Materials and Methods and we have added the following edits.

Using the three data sources (i.e. contact data, de-identified manifest and de-identified location records), we categorised the contacts between each dyad into one of four contact groups, g , namely (i) passenger-passenger contact from within the same travelling group (i.e. passengers in the same cabin or having a cumulative contact duration of more than 5 hours over three days), (ii) passenger-passenger contact from different travelling groups, (iii) crew-crew contact, and (vi) passenger-crew contact. 5 hours was selected as a conservative definition for travelling groups, given that this is considerably longer than an average meal duration and more than 99% of the cumulative contact duration (i.e. sum of all contact episodes) between passengers from different cabins were less than this duration.

All in all, I think it is worth to look at the appendix from the perspective of a reader and explain the model for her.

Summary:

The article is interesting in my opinion, but I have the impression that there are some technical issues that should be addressed. Furthermore, while the results might be interesting for organizers of cruises, I’m not sure what we can learn for the overall control of the delta variant in daily life. And, cruises are not known to be important factors for the spread of corona.

All in all, I’m not sure if the article is suited for a journal as nature communications addressing a general audience; perhaps a more specialized journal would be better suited.

Thank you again for your comments and we agreed that addressing each of these points has helped improve the clarity of the manuscript. As this work holds broad implications for predicting the spread of infectious disease on empirical networks under various simulated contagion-control interventions (as we show throughout the paper), we believe that the revisions made in response to all reviewers will increase the manuscript’s relevance as compared to the first draft.

REVIEWERS' COMMENTS

Reviewer #2 (Remarks to the Author):

Thanks for the revised version - in my opinion, the authors carefully addressed all comments of the reviewers.

The data set their model is based on and the findings are for sure interesting; however, I'm still not persuaded that the results can be carried over to more general situations. Anyhow, given that the editor comes to the conclusion that modelling of cruises is interesting for Nat. Com., I'm fine with the revised manuscript.